# Methylated Cell-Free Tumor DNA in Sputum as a Tool for Diagnosing Lung Cancer—A Systematic Review and Meta-Analysis

**DOI:** 10.3390/cancers16030506

**Published:** 2024-01-24

**Authors:** Sara Witting Christensen Wen, Morten Borg, Signe Timm, Torben Frøstrup Hansen, Ole Hilberg, Rikke Fredslund Andersen

**Affiliations:** 1Department of Oncology, Vejle Hospital, University Hospital of Southern Denmark, 7100 Vejle, Denmark; 2Department of Regional Health Research, University of Southern Denmark, 5000 Odense, Denmark; 3Department of Medicine, Vejle Hospital, University Hospital of Southern Denmark, 7100 Vejle, Denmark; morten.hornemann.borg@rsyd.dk; 4Department of Biochemistry and Immunology, Vejle Hospital, University Hospital of Southern Denmark, 7100 Vejle, Denmark

**Keywords:** lung cancer, sputum, tumor DNA, DNA methylation, liquid biopsy

## Abstract

**Simple Summary:**

Lung cancer is one of the deadliest cancers worldwide, and the prognosis is poor. The disease is potentially curable if detected at an early stage, but currently, the only widely implemented screening tool is low-dose computed tomography. Biomarkers such as methylated tumor DNA may be used for the early detection of lung cancer, and sputum is an appealing, non-invasive sample type. With this systematic review and meta-analysis, we aimed to identify all studies evaluating the quantitative methylation of tumor DNA in sputum samples for lung cancer detection. A systematic overview of all the available evidence may highlight areas of improvement as well as certain high-performing genes to prioritize in future studies.

**Abstract:**

Lung cancer is the leading cause of cancer-related mortality worldwide. Early diagnosis is pivotal for the prognosis. There is a notable overlap between lung cancer and chronic bronchitis, and the potential use of methylated tumor DNA in sputum as a biomarker for lung cancer detection is appealing. This systematic review and meta-analysis followed the PRISMA 2020 statement. A comprehensive search was conducted in Embase, Medline, Web of Science, and the Cochrane Library, using these search strings: Lung cancer, sputum, and methylated tumor DNA. A total of 15 studies met the eligibility criteria. Studies predominantly utilized a case–control design, with sensitivity ranging from 10 to 93% and specificity from 8 to 100%. A meta-analysis of all genes across studies resulted in a summary sensitivity of 54.3% (95% CI 49.4–59.2%) and specificity of 79.7% (95% CI 75.0–83.7%). Notably, two less explored genes (TAC1, SOX17) demonstrated sensitivity levels surpassing 85%. The study’s findings highlight substantial variations in the sensitivity and specificity of methylated tumor DNA in sputum for lung cancer detection. Challenges in reproducibility could stem from differences in tumor site, sample acquisition, extraction methods, and methylation measurement techniques. This meta-analysis provides a foundation for prioritizing high-performing genes, calling for a standardization and refinement of methodologies before potential application in clinical trials.

## 1. Introduction

Lung cancer is a global health challenge and the leading cause of cancer death world-wide [1]. Diagnosis at an early stage is pivotal for prognosis [2]. Unfortunately, a substantial portion of lung cancer cases receive a late-stage diagnosis, limiting curative treatment options [3]. This primarily stems from the fact that the majority of early-stage lung cancer is incidentally discovered [4], as symptoms typically manifest only in advanced stages [5].

Computed tomography (CT) screening has emerged as a valuable tool for the early detection of lung cancer and has been implemented in various countries, leading to a shift in diagnosing lung cancer at earlier stages [6,7,8,9]. However, CT screening is a costly and time-consuming process for radiologists [10], creating a demand for a biomarker capable of detecting lung cancer or identifying high-risk individuals [11].

In many countries, there is a notable overlap between lung cancer and chronic bronchitis [12], a condition marked by daily sputum production. Previous attempts at lung cancer screening using sputum cytology demonstrated low sensitivity [13], and it is only favored in specific countries [14]. Assessing the presence of methylated tumor DNA in sputum has demonstrated potential as a more efficient substitute for conventional cytology techniques [15]. Methylated tumor DNA can be quantified using various molecular techniques, often relying on polymerase chain reaction (PCR) methods like real-time PCR, quantitative methylation-specific PCR (QMSP), and digital PCR [16]. Additional approaches include pyrosequencing and next-generation sequencing (NGS) methods [17].

Over the years, multiple studies have evaluated the diagnostic accuracy of methylated tumor DNA in the sputum of a large number of genes for lung cancer detection [18,19,20]. These studies have provided considerable variation in sensitivity and specificity, possibly due to differences in sputum sample collection methods. Furthermore, the method used to analyze methylated tumor DNA can profoundly impact the results, emphasizing the importance of clearly reporting the methodology for result reproducibility and comparison. An up-to-date systematic review and meta-analysis of the current evidence are necessary for a more comprehensive understanding of the diagnostic performance of methylated DNA in sputum.

The primary objective of this systematic review and meta-analysis on methylated tumor DNA in sputum for lung cancer diagnosis was to systematically identify and analyze all relevant studies investigating the diagnostic accuracy of this approach.

## 2. Materials and Methods

### 2.1. Study Protocol and Registration

The present systematic review was conducted according to the Preferred Reporting Items for Systematic Reviews and Meta-Analyses (PRISMA) 2020 statement [21]. The study protocol was registered in the PROSPERO database (registration number CRD42023453066, University of York, York, UK) on 21 August 2023 [22].

### 2.2. Data Sources and Search Strategy

Systematic literature searches were performed in the online databases Embase and Embase Classic, Ovid Medline, Web of Science, and Cochrane Library. The search strings consisted of three main concepts divided into search blocks: (A) lung cancer, (B) sputum, and (C) methylated tumor DNA. Each block contained the relevant Subject Headings (Embase and Medline) or Medical Subject Headings (Cochrane Library) and free-text keywords. All keywords were included by incorporating the Boolean operator “OR”. The three blocks were then combined using the Boolean operator “AND”, and the resulting references were retrieved for each database on 21 August 2023. We conducted citation searches (forward and backward) on all studies included after full-text evaluation using the Web of Science database. These references were extracted on 3 October 2023.

The searches were not limited by language, article type, or year of publication. The search strings were constructed by the authors assisted by a research librarian, and the search strings for each database are available in the Appendix A.

### 2.3. Reference Screening and Eligibility Criteria

All references were imported into the web-based systematic review platform Covidence (Covidence, Melbourne, Australia). Duplicates were automatically identified and removed by Covidence. All references were screened by title and abstract by two independent reviewers (MB and SW), and consensus was reached by discussion with the option to involve a third reviewer (RA) in case of persistent disagreement. All potentially relevant studies were reviewed in full by two independent reviewers (MB and SW) and scored according to the following inclusion criteria: (A) adult patients with a diagnosis of lung cancer or patients undergoing diagnostic work-up or screening for lung cancer, and adult healthy control subjects or non-cancer control patients; (B) spontaneous or induced sputum collected for analysis of methylated tumor DNA using a quantitative analysis method; (C) tumor cytology or histopathology used as the reference standard; (D) diagnostic performance of the biomarker(s) reported as contingency data with sufficient information to calculate sensitivity and specificity. The exclusion criteria comprised (A) case reports, meeting abstracts, editorials, comments, notes, letters, and literature reviews; (B) studies in languages other than English; (C) in silico analyses of public data repositories.

### 2.4. Data Extraction and Quality Assessment

The data extraction form was pilot-tested in Covidence with three studies. Basic study characteristics included the study ID (the first author’s last name and the year of publication), geographic region, study design, number and description of lung cancer patients including histology and stage, number and description of control subjects, and the choice of reference standard. Based on previous experience, we included an assessment of how the Methods section was reported inspired by the Minimum Information for publication of Quantitative real-time PCR Experiments (MIQE) guidelines [23]. These items included sample collection, which part of the sputum sample was used for analysis, DNA extraction, primer and probe sequences, reaction volume and amount of DNA, thermocycling parameters, assay type, calibration curves, and diagnostic cutoff. Extracted outcomes included gene name(s), diagnostic sensitivity and specificity, and the number of true positives, false negatives, true negatives, and false positives in a contingency table.

The overall study quality was evaluated according to the Quality Assessment of Diagnostic Accuracy Studies 2 (QUADAS-2) tool, which contains four domains covering patient selection, index test, reference standard, and flow and timing [24]. We also included study funding and author conflicts of interest.

All data extraction and quality assessments were carried out independently and blindly by two reviewers (SW and MB), and disputes were settled by discussion. In case of methodology specific issues, we consulted an expert (RA), who had the deciding vote.

### 2.5. Statistical Analysis

Study characteristics and selected MIQE items were collated in summary tables. Diagnostic sensitivity was calculated by the following formula: True positive/(true positive + false negative). Diagnostic specificity was calculated as follows: True negative/(true negative + false positive). A diagnostic test meta-analysis was carried out on all studies with sufficient data. The summary effect estimates were derived from the STATA command metandi, which was also used for the hierarchical summary receiver operating characteristics (HSROC) plot [25]. This procedure consists of a two-level mixed-effect logistic regression model based on an independent binomial distribution. Single-gene performances across the included studies were illustrated by forest plots of sensitivity and specificity using the STATA command midas [26]. Deek’s funnel plot and Deek’s funnel plot asymmetry test were used for evaluating the risk of publication bias with a level of significance set at 0.05. All analyses were performed in STATA BE version 18 (StataCorp LLC, College Station, TX, USA), and forest plot graphics were executed in Microsoft Excel (Microsoft Corporation, Redmond, WA, USA).

## 3. Results

### 3.1. Search Results and Study Selection

The results of the database searches and the subsequent selection process are illustrated in Figure 1. The initial searches, carried out in Embase, Medline, Web of Science, and Cochrane Library, identified 1299 potentially eligible records. Forward and backward citation searches of the 15 included studies resulted in 1154 additional references, many of which were duplicates. A total of 1234 references were eliminated based on title and abstract screening, which left 67 studies eligible for full-text assessment. Eventually, 15 studies were included in the review.

### 3.2. Study Characteristics

The included studies were published between 2007 and 2021 (Table 1 and Appendix A). The majority of the studies (9/15) originated from European countries, 3/15 were from North America, and 3/15 were from Asia (Table 1). Only two of the studies were cohort studies [27,28], while the remaining were of a case–control design. Four studies presented both a training and a validation cohort [18,19,29,30]. The number of cases and controls ranged from 13 to 159 and 24 to 159, respectively, with a median number of 56 cases and 68 controls. The reference standard was histology or cytology in 9/19 cohorts, histopathology of a surgery specimen alone or in combination with a tissue biopsy in 2/19 and 2/19 cohorts, respectively, while the reference standard was not described for 6/19 cohorts.

### 3.3. Biological Sample Types and Collection Method

Studies analyzing both spontaneously produced sputum and induced sputum were included in the present review. Spontaneous sputum constituted the main sample type in the studies (11/15), induced sputum was used in 3/15 studies, and the specific induction method Lung Flute was used in 1/15 studies. The Lung Flute used in the study by Li et al. is a handheld therapy device employing positive expiratory pressure in combination with sound waves to mobilize mucus in the airways [30,39]. Induced sputum is produced by letting the patient inhale a nebulized hypertonic saline solution to help sputum production [40].

Only 3/15 studies used a single-day sample, 11/15 used pooled sputum samples from multiple days, and one study did not describe the timing of sputum collection. The specific part of the sputum sample used for methylated DNA analysis was generally not described (11/15), but 3/15 studies reported using the cell pellet, and one study performed a comparison of pellet and supernatant. Detailed information for each study can be viewed in Appendix A.

### 3.4. The Reporting of Key Domains of the Analysis Methods

Various methods were applied for methylated tumor DNA analysis by the included studies, but they were mainly PCR-based, and only one study applied a sequencing approach [35]. We identified eight key domains from the MIQE guidelines to aid in assessing how well the methods were reported by the included studies [23]. These eight domains were assessed in a simplified manner by a score of either “Yes” (reported) or “No” (not reported) for each item, and the results are presented in Table 2. Overall, the reporting quality was good, yet only 5/15 studies reported adequate information for all eight domains [27,30,31,34,37]. The remaining 10/15 studies failed to report information on at least one key domain. One study did report most of the required information; however, the information was reported in multiple references and it was therefore difficult to obtain with certainty [29]. More details regarding DNA extraction kits and the amount of DNA used can be found in Appendix A. 

### 3.5. Methylated DNA Analysis in Sputum for the Diagnosis of Lung Cancer

A total number of 31 genes were investigated in the 15 included studies with frequencies ranging from 1 (17 genes) to 13 (1 gene, RASSF1A). The comprehensive list of genes is available in the Appendix A. Besides RASSF1A, APC and CYGB were the most frequent genes targeted in nine and eight cohorts, respectively. The diagnostic sensitivity reported by the studies varied and ranged from 10 to 93%, and the specificity had a similar range of 8–100%.

The sensitivity of aberrantly methylated RASSF1A ranged from 17 to 57% as reported in the 13 independent cohorts investigating this gene with a pooled sensitivity estimate of 39% (95% CI 14–68%). The individual results are presented in Figure 2a. The diagnostic specificity of RASSF1A was generally high with a range of 75–100% and a pooled estimate of 94% (95% CI 90–96%) (Figure 2b).

As the second most frequently investigated gene, APC had a diagnostic sensitivity of 17–63% in nine independent cohorts and a pooled sensitivity estimate of 44% (95% CI 32–56%) (Figure 3a). The specificity ranged from 55 to 100% with a pooled effect estimate of 82% (95% CI 66–92%) (Figure 3b).

The third most frequently analyzed gene in eight cohorts was CYGB with a diagnostic sensitivity ranging from 31 to 57% with a summary estimate of 47% (95% CI 40–54%) and a specificity ranging from 46 to 100% with a summary estimate of 82% (95% CI 64–92%) (Figure 4a,b).

A meta-analysis of the comprehensive range of genes analyzed in all independent cohorts resulted in 87 discrete data points and summary sensitivity and specificity estimates of 54.3% (95% CI 49.4–59.2%) and 79.7% (95% CI 75.0–83.7%), respectively. The corresponding diagnostic odds ratio was 4.7 (95% CI 3.8–5.7). The area under the HSROC curve was 0.71 (95% CI 0.67–0.75), and the 95% confidence region as well as the 95% prediction region are illustrated in Figure 5. Two of the less frequently investigated genes, SOX17 and TAC1, showed a high diagnostic sensitivity >85% with a corresponding specificity of >70% in three studies [28,30,38]. The comprehensive contingency data can be accessed in Appendix A, and the HSROC graph with an ID for each data point can be viewed in Appendix A.

### 3.6. Quality Assessment and Risk of Bias

We used the QUADAS-2 checklist to evaluate the risk of bias in the included studies as recommended for test accuracy studies by the *Cochrane Handbook for Systematic Reviews of Interventions* [41]. The studies were of an acceptable quality overall, but issues in the “Patient selection” domain were frequent (Figure 6). Only two studies received “Low risk” in all seven judgments across the four domains [28,30]. The comprehensive quality assessments are reported in the Appendix A.

We evaluated the potential risk of small study effects/publication bias using Deek’s funnel plot asymmetry test (*p* = 0.096) and the corresponding graph (Appendix A) and found no statistically significant risk of publication bias.

## 4. Discussion

This systematic review and meta-analysis aimed to evaluate the quantitative detection of methylated tumor DNA in sputum samples as a diagnostic tool in lung cancer patients. A total of 15 studies met the eligibility criteria for inclusion in this review. The studies displayed varying sensitivity and specificity in the detection of methylated tumor DNA, with a summary sensitivity estimate of 54.3% (95% CI 49.4–59.2%) and a summary specificity estimate of 79.7% (95% CI 75.0–83.7%). The area under the HSROC curve was 0.71 (95% CI 0.67–0.75).

The use of tumor DNA methylation has been proposed for screening applications in both blood [42] and sputum [43]. Sputum analysis as a screening approach for detecting lung cancer is appealing. It is non-invasive and bears no significant risk or inconveniences for the participants. Nevertheless, past research has indicated that cytological evaluation of sputum, even when combined with concurrent chest X-ray assistance, may not be sufficient for effective screening [44]. The current meta-analysis examining the detection of lung cancer through the analysis of methylated tumor DNA in sputum reveals considerable variations in sensitivity, strongly influenced by the specific gene under investigation. Even when focusing on the three most extensively studied genes (RASSF1A, APC, and CYGB), variations persist, underscoring the overall challenges in reproducibility. It is imperative to scrutinize factors such as the site and stage of lung cancer, sputum production method, DNA extraction technique, and analysis methodology. Each of these steps holds equal importance in assessing the viability of a particular gene as a feasible screening target for clinical application.

Most frequently, the studies employed a case–control design (12 out of 15 studies). This design confers distinct advantages, allowing for the execution of retrospective studies using readily available sputum samples and the possibility of matching the controls and the cases based on various parameters. The retrospective study design, however, entails a higher risk of introducing bias. Although prospective cohort studies entail greater costs and time investment, they offer a superior level of evidence [45]. No randomized clinical trials were identified.

The objective of lung cancer screening is to identify early-stage cases amenable to curative treatment options [46]. However, a significant portion of stage I/II lung cancer tumors [47] may not be in close proximity to large bronchi, and sputum production may not originate from the tumor site. Unfortunately, the large majority of studies lacked the provision of sensitivity and specificity specific to stage, histological type, or central/peripheral location of lung cancer. Consequently, the creation of a meta-analysis categorized by stage, histology, or location was not possible. Additionally, the predominant use of spontaneous sputum samples may not adequately represent deep bronchial origins [48]. These factors potentially contribute to the relatively low sensitivity observed in the studies.

Another important factor is the pre-analytical handling of the sputum samples. Optimal pre-analytical conditions are pivotal for the results when analyzing tumor DNA in blood samples [49]. To our knowledge, no standardized guidelines exist for the optimal pre-analytical handling of sputum samples intended for tumor DNA analysis, but some of the recommendations regarding circulating tumor DNA may reasonably be applied to sputum analysis. Methylated tumor DNA is present in sputum in very limited quantities, and the final amount retrieved from the sample may vary depending on the chosen collection method and the part of the sample used for analysis. Only the study by van der Drift et al. from 2008 compared supernatant and pellet; they found that RASSF1A was methylated in 20% and 42.9% of the lung cancer patients for cell-free and cellular DNA, respectively [33]. This indicates that the cell pellet may achieve better sensitivity. However, as this was only reported in one study, further investigation is needed.

The sample storage time can also affect the DNA yield as shown in a study by Sozzi et al. who estimated a yearly DNA decay rate of around 30% for both plasma and purified DNA frozen at −80 °C [50]. The study by Leng et al. utilized a patient cohort from a study initiated in 1993 and the study was published in 2012, resulting in a potential storage time of up to 19 years [19]. Several other studies also reported potentially extensive storage times of 10 years or more [28,36,37]. The kits or methods used for DNA extraction should always be reported in studies investigating methylated tumor DNA, as stated in the MIQE guidelines, as the DNA yield can vary depending on the kit, and only two studies failed to report these data [23]. Most of the included studies used DNA extraction kits intended for blood, which are optimized for longer DNA fragments. This might affect the diagnostic sensitivity, since tumor DNA tends to be more fragmented than normal DNA [51].

Variations in results may also arise from differences in the analysis of sputum methylation. An inclusion criterion for this study was the adoption of quantitative measurements of methylated DNA. Among the selected studies, the majority (12 out of 15) employed QMSP, while 2 out of 15 utilized digital PCR, and 1 out of 15 employed pyrosequencing. The prevalence of QMSP may be attributed to the fact that it is an older and very well-established technology, and the studies in this review date from 2007 and onward. Digital PCR is a more current technique, which was used by two studies in the current review [30,38]. Acquiring a digital PCR platform is costly, and quantitative PCR can produce high-quality, quantitative results if the right guidelines for sample processing and data analysis are followed [52]. Previously, both QMSP and digital PCR using MethyLight have demonstrated a robust correlation between expected and observed methylation values [16]. However, digital PCR exhibits higher sensitivity, as illustrated previously, with a 20-fold lower detection limit with the droplet digital PCR technique compared to standard quantitative PCR [53]. The study by Su et al. from 2018 compared droplet digital PCR with QMSP and found a lower limit of quantification for the digital PCR approach, albeit not a 20-fold difference as discussed above [38]. An extensive interlaboratory study found that droplet digital PCR could achieve highly reproducible absolute quantification of a specific DNA target, with an inter-laboratory difference of less than 12% [54]. Taken together, digital PCR seems a favorable method for analyzing methylated tumor DNA.

Methylation analysis in sputum samples is generally not very widely investigated, and we only identified 15 studies for the present review as opposed to 33 studies evaluating the use of methylated tumor DNA in blood samples for lung cancer detection [42]. Sputum may be considered a material with limited potential for this purpose, since only one study was published in the period from 2020 to 2022, whereas 13 of the 33 studies of methylated tumor DNA in blood were published in the same time period. Blood plasma is a more standardized material compared to sputum, and there are several DNA extraction kits aimed specifically at cell-free DNA, which is not the case for sputum. Not all patients are able to produce a sputum sample, while blood can be sampled in almost any situation. As a result, plasma may be a better biological sample type than sputum for analyzing minimally invasive biomarkers. However, sputum, bronchial lavage, and other sample types collected in closer proximity to the tumor might be relevant in specific situations.

While most genes in the present review exhibited a relatively low sensitivity of around 50%, a subset of less explored genes demonstrated sensitivity levels surpassing 85%, with specificity consistently above 70%. Notably, these genes (TAC1, SOX17; Appendix A) were all investigated in the same three studies, which were three of the most recently published, and two of them used digital PCR [28,30,38]. It is also worth noting that two of these studies were the only ones to receive a score of “Low risk” across all four QUADAS-2 domains [28,30]. Future research on methylated sputum DNA could advantageously prioritize these genes and focus on optimizing and standardizing sample types, extraction methods, and methylation analysis techniques. The diagnostic potential may be further improved by incorporating other types of biomarkers such as miRNA, as performed by Li 2021 [30] and Su 2016 [20], or as a combination of biomarkers and other characteristics [55].

Our search strategy employed broad criteria, scanning major databases to identify a substantial number of potentially relevant studies. Nevertheless, we acknowledge the possibility of not capturing all pertinent research, as some studies may have used gene names without broader ctDNA terms in their titles or abstracts, resulting in their exclusion from our search results. Additionally, the review did not encompass gray literature, congress abstracts, unpublished results, or studies in a language other than English.

Pooling data from multiple studies in this review and meta-analysis enhances sample size and statistical power, thereby improving the precision and reliability of the findings. This approach supplies a comprehensive synthesis of the available evidence on the diagnostic accuracy of methylated DNA in sputum of lung cancer, facilitating the identification of possible patterns or tendencies across studies. However, it is imperative to acknowledge that the results may not be universally applicable to all populations, given differences in the race and smoking prevalence of lung cancer patients. This caveat is particularly noteworthy as methylation has been established to be heavily influenced by smoking [56]. Thus, caution is advised in interpreting the summary estimates, considering the extensive and diverse array of investigated genes and cohorts.

The included studies were of varying quality, with potential limitations or biases in design, conduct, or reporting affecting the validity and reliability of the results. The predominance of case–control studies with healthy subjects as the control group introduces spectrum bias, emphasizing the need for more accurate effect estimates from cohort studies or case–control studies involving patients with benign diseases. The risk of publication bias was considered; however, Deek’s funnel plot asymmetry test did not indicate a significant concern. Heterogeneity across studies, stemming from variations in case and control populations, analysis methods, cutoff values, and study designs, introduces a level of uncertainty in the generalizability of the findings. It is crucial to recognize these limitations and exercise caution when interpreting and applying the results of this meta-analysis.

## 5. Conclusions

In summary, the accuracy in identifying lung cancer through the assessment of methylated tumor DNA in sputum exhibited notable discrepancies among various studies. These variations are believed to stem from differences in tumor location, lung cancer stage, sample procurement, DNA extraction methodologies, and approaches to measuring methylation. With an overall sensitivity estimate of 55% (specificity at 80%), it is evident that improvements are required before contemplating inclusion in a randomized controlled trial. Uniform and consistent reporting of materials and methods are essential for the interpretation and reproducibility of results, and this research area would benefit from standardization. Nonetheless, this meta-analysis establishes a foundation for pinpointing genes with comparatively higher sensitivity and specificity, urging further exploration with a focus on refining the mentioned factors.

## Figures and Tables

**Figure 1 cancers-16-00506-f001:**
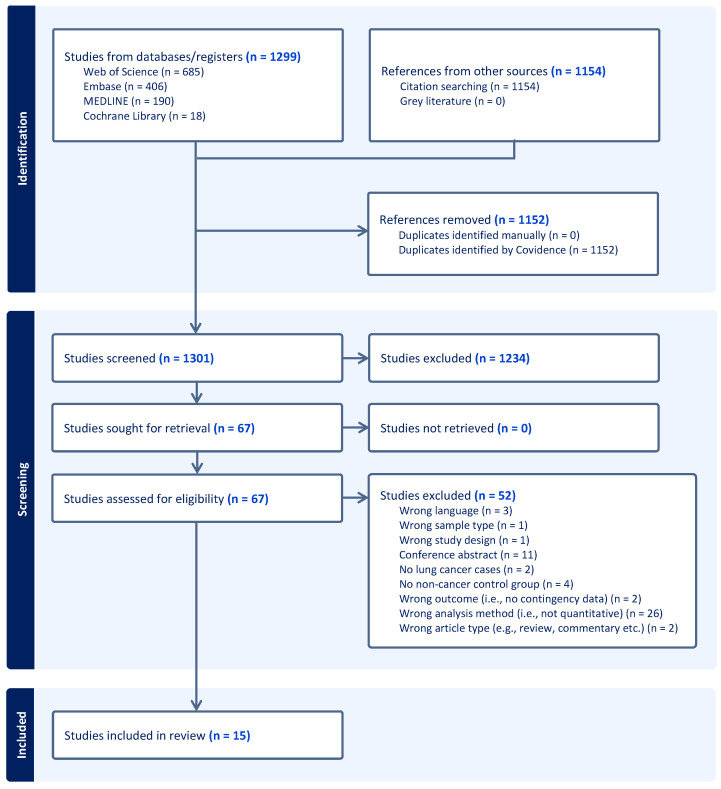
PRISMA flowchart illustrating the study selection process.

**Figure 2 cancers-16-00506-f002:**
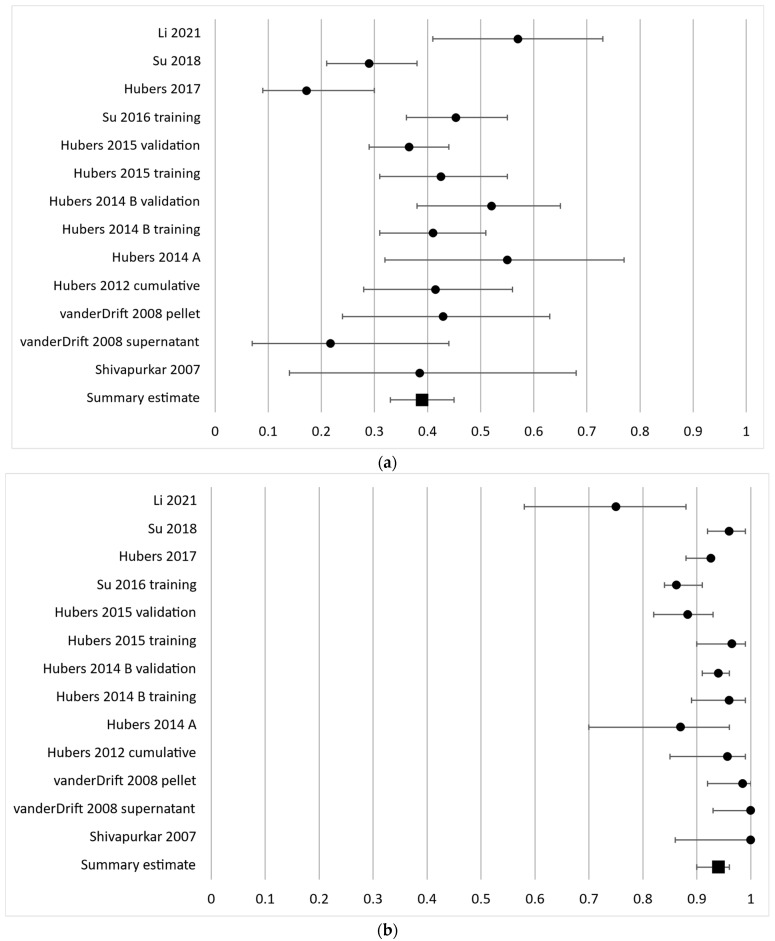
Forest plot of sensitivity (**a**) and specificity (**b**) for the most frequently analyzed gene, RASSF1A. The black circles represent the diagnostic performance estimates, and the error bars represent the calculated 95% confidence intervals. The black square represents the summary estimate for all independent cohorts [18,20,27,29,30,31,33,36,37,38].

**Figure 3 cancers-16-00506-f003:**
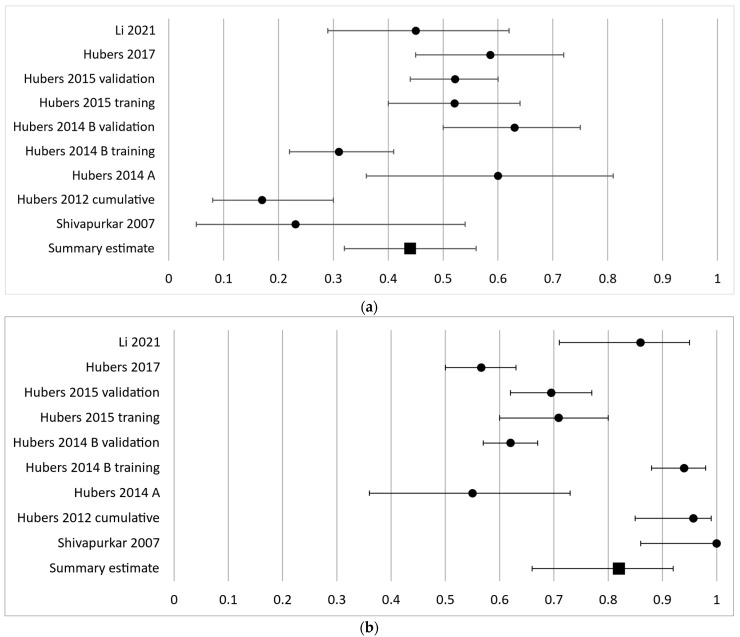
Forest plot of sensitivity (**a**) and specificity (**b**) for the second most frequently analyzed gene, APC. The black circles represent the diagnostic performance estimates, and the error bars represent the calculated 95% confidence intervals. The black square represents the summary estimate for all independent cohorts [18,27,29,30,31,36,37].

**Figure 4 cancers-16-00506-f004:**
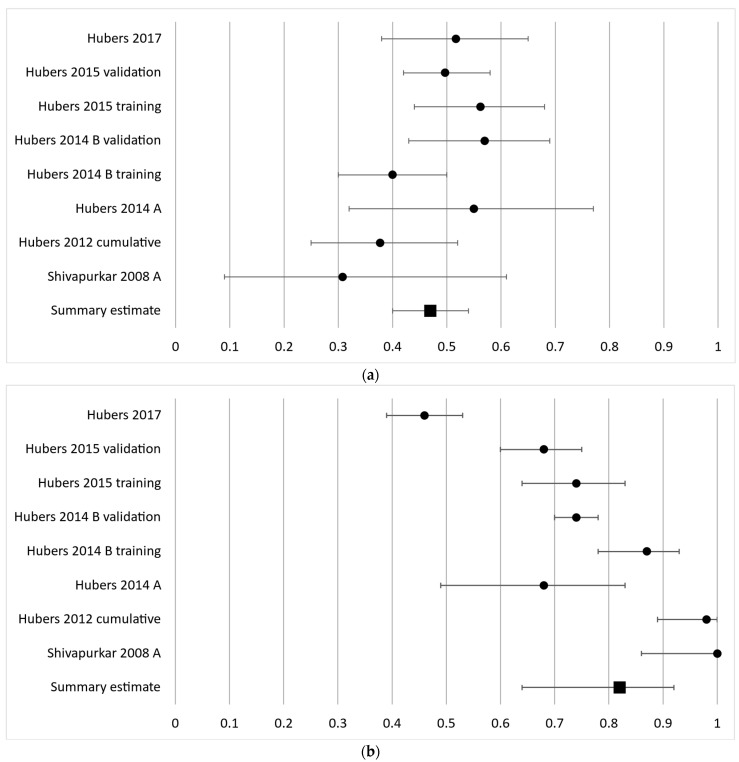
Forest plot of sensitivity (**a**) and specificity (**b**) for the third most frequently analyzed gene, CYGB. The black circles represent the diagnostic performance estimates, and the error bars represent the calculated 95% confidence intervals. The black square represents the summary estimate for all independent cohorts [18,27,29,32,36,37].

**Figure 5 cancers-16-00506-f005:**
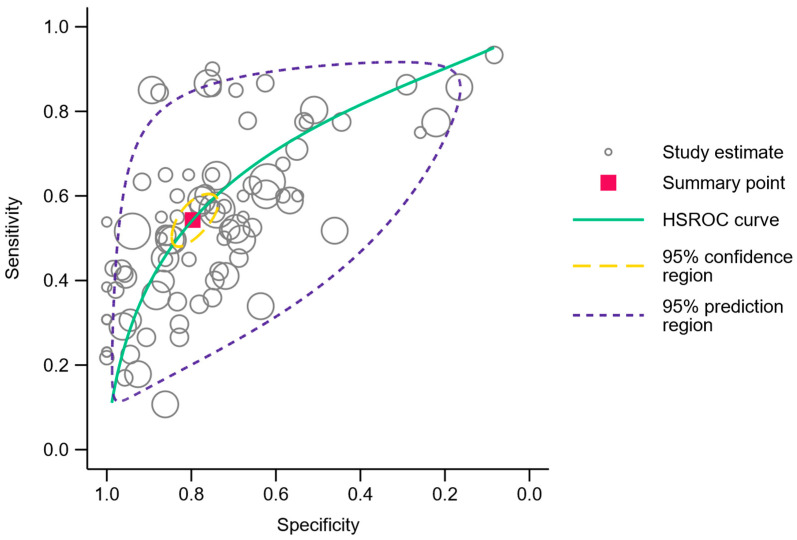
Hierarchical summary receiver operating characteristics plot. Each open circle represents a gene analyzed in an independent cohort. The summary point is represented by the red square, and the 95% confidence region and 95% prediction region are outlined in dashed yellow and purple lines, respectively.

**Figure 6 cancers-16-00506-f006:**
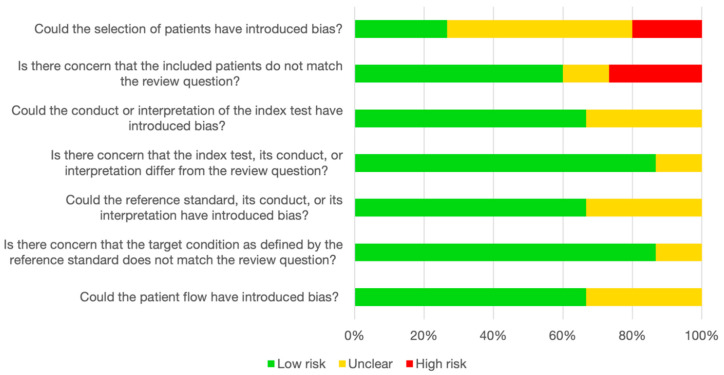
Assessment of study quality according to QUADAS-2. The stacked bar chart represents the consensus judgments in each of the four domains covered by QUADAS-2. Green: low risk of bias/low level of concern. Yellow: unclear risk of bias/unclear level of concern. Red: high risk of bias/high level of concern. QUADAS-2, Quality Assessment of Diagnostic Accuracy Studies 2.

**Table 1 cancers-16-00506-t001:** Study characteristics for all independent cohorts included. Stage I-IV as reported in the studies. LUAD, lung adenocarcinoma; LUSC, lung squamous cell carcinoma; SCLC, small-cell lung cancer.

Study ID	Region	Study Design	Cases	Histology	Stage	Controls	Cases (n)	Controls (n)	Reference Standard
Shivapurkar, 2007 [31]	Europe	Case–control study	Retrospectively selected patients with lung cancer	LUSC 8/13 (62%), LUAD 5/13 (38%)	I 2/13 (15%), II 2/13 (15%), III 4/13 (31%), IV 4/13 (31%), unknown 1/13 (8%)	Unmatched controls with benign diseases; other: four patients with prior lung cancer.	13	25	Histology or cytology not specified
Shivapurkar, 2008 A [32]	Europe	Case–control study	Retrospectively selected patients with lung cancer	Unknown 13/13 (100%)	Unknown 13/13 (100%)	Unmatched controls with benign diseases	13	25	Not described
van der Drift, 2008 [33]	Europe	Case–control study	Retrospectively selected patients with lung cancer	LUSC 13/28 (46%), LUAD 9/28 (32%), SCLC 4/28 (14%), other 2/28 (7%%)	I 4/28 (14%), II 5/28 (18%), III 4/28 (14%), IV 6/28 (21%), unknown 9/28 (32%)	Unmatched controls with benign diseases	28	68	Histology or cytology not specified
Shivapurkar, 2008 B [34]	Europe	Case–control study	Retrospectively selected patients with lung cancer	Unknown 13/13 (100%)	Unknown 13/13 (100%)	Unmatched controls with benign diseases	13	25	Not described
Hwang, 2011 [35]	Asia	Case–control study	Retrospectively selected patients with lung cancer	LUSC 34/76 (45%), LUAD 42/76 (55%)	I 14/76 (18%), II 5/76 (7%), III 28/76 (37%), IV 29/76 (38%)	Unmatched, healthy controls; unmatched controls with benign diseases	76	109	Histology or cytology not specified
Hubers, 2012 [27]	Europe	Cohort study	Lung cancer cases from a cohort study	Unknown 53/53 (100%)	Unknown 53/53 (100%)	Non-cancer participants from a cohort study	53	47	Not described
Leng, 2012 [19]	North America	Case–control study	Retrospectively selected patients with lung cancer	Cohort 1: unknown 64/64 (100%).	Cohort 1: unknown 64/64 (100%).	Matched on certain characteristics	Cohort 1: 64.	Cohort 1: 64.	Not described
				Cohort 2: unknown 40/40 (100%)	Cohort 2: Stage I 40/40 (100%)		Cohort 2: 40	Cohort 2: 90.	Histopathology of surgery specimen
Hubers, 2014 A [36]	Europe	Case–control study	Retrospectively selected patients with lung cancer	LUSC 6/20 (30%), LUAD 7/20 (35%), SCLC 1/20 (5%), other 6/20 (30%)	I 1/20 (5%), II 3/20 (15%), III 9/20 (45%), IV 7/20 (35%)	Unmatched controls with benign diseases	20	31	Histology or cytology not specified
Hubers, 2014 B [29]	Europe	Case–control study	Retrospectively selected patients with lung cancer	Cohort 1: unknown 98/98 (100%).	Cohort 1: unknown 98/98 (100%).	Matched on certain characteristics	Cohort 1: 98.	Cohort 1: 90.	Not described
				Cohort 2: Unknown 60/60 (100%)	Cohort 2: 60/60 (100%)		Cohort 2: 60	Cohort 2: 445.	Not described
Hubers, 2015 [18]	Europe	Case–control study	Lung cancer patients at diagnosis but also at progression on treatment	Cohort 1: LUSC 31/73 (42%), LUAD 26/73 (36%), SCLC 1/73 (1%), other 15/73 (21%).	Cohort 1: I 14/73 (19%), II 9/73 (12%), III 24/73 (33%), IV 25/73 (34%), unknown 1/73 (1%).	Unmatched controls with benign diseases; patients with benign diseases; patients who had surgery for lung cancer and remained cancer-free for 3 years.	Cohort 1: 73.	Cohort 1: 86.	Histology or cytology not specified
				Cohort 2: LUSC 50/159 (31%), LUAD 66/159 (42%), SCLC 6/159 (1%), other 37/159 (23%)	Cohort 2: I 29/159 (18%), II 17/159 (11%), III 47/159 (30%), IV 66/159 (42%)		Cohort 2: 159	Cohort 2: 154.	Histology or cytology not specified
Su, 2016 [20]	Asia	Case–control study	Retrospectively selected patients with lung cancer	LUSC 54/117 (46%), LUAD 63/117 (54%)	I 117/117 (100%)	Matched on certain characteristics	117	174	Histopathology of tissue biopsy or surgery specimen
Hubers, 2017 [37]	Europe	Case–control study	Lung cancer cohort from another study	LUSC 7/56 (13%), LUAD 34/56 (61%), SCLC 2/56 (4%), other 8/56 (14%), unknown 5/56 (9%)	I 36/56 (64%), II 4/56 (7%), III 6/56 (11%), IV 10/56 (18%)	Non-cancer participants from a cohort study	56	217	Histopathology of tissue biopsy; cytology
Hulbert, 2017 [28]	North America	Cohort study	Lung cancer cases from a cohort study	Unknown 90/90 (100%)	Unknown 90/90 (100%)	Non-cancer participants from a cohort study	90	24	Histopathology of surgery specimen
Su, 2018 [38]	Asia	Case–control study	Retrospectively selected patients with lung cancer	LUSC 57/127 (45%), LUAD 63/127 (50%), other 7/127 (6%)	I 33/127 (26%), II 32/127 (25%), III 29/127 (23%), IV 33/127 (26%)	Matched on certain characteristics	127	159	Histopathology of tissue biopsy or surgery specimen
Li, 2021 [30]	North America	Cohort study	Lung cancer cases from a cohort study	Cohort 1: LUSC 18/40 (45%), LUAD 22/40 (55%).	Cohort 1: I 13/40 (33%), II 13/40 (33%), III–IV 14/40 (35%).	Non-cancer participants from a cohort study	Cohort 1: 40.	Cohort 1: 36.	Histology or cytology not specified
				Cohort 2: LUSC 16/36 (44%), LUAD 20/36 (56%)	Cohort 2: I 13/36 (36%), II 12/36 (33%), III–IV 11/36 (31%).		Cohort 2: 36.	Cohort 2: 39.	Histology or cytology not specified

**Table 2 cancers-16-00506-t002:** Reporting of PCR-based methods. N/A, not applicable; PCR, polymerase chain reaction; QMSP, quantitative methylation-specific PCR; ref, reference.

Study ID	Was the DNA Extraction Kit Name Reported?	Analysis Method	Were the Primer Sequences Reported?	Were the Probe Sequences Reported?	Were the Reaction Volume and Amount of DNA Reported?	Were the Complete Thermocycling Parameters Reported?	Assay Type	Were the Calibration Curves or Serial Dilutions Reported?	How Was the Cutoff Determined?
Shivapurkar, 2007 [31]	Yes	QMSP	Yes	Yes	Yes	Yes	Singleplex	Yes	Defined by a training cohort (unvalidated)
Shivapurkar, 2008 A [32]	Yes	QMSP	Yes	Yes	Yes	Yes	Singleplex	No	Defined in a previous study
van der Drift, 2008 [33]	Yes	QMSP	Yes	Yes	Yes	Yes	Singleplex	No	Not described
Shivapurkar, 2008 B [34]	Yes	QMSP	Yes	Yes	Yes	Yes	Singleplex	No	Defined in a previous study
Hwang, 2011 [35]	No	Sequencing	Yes	N/A	Yes	Yes	Singleplex	N/A	Not described
Hubers, 2012 [27]	Yes	QMSP	Yes	Yes	Yes	Yes	Multiplex	Yes	Arbitrarily set at a specific level of sensitivity or specificity
Leng, 2012 [19]	Yes	QMSP	Yes	No	No	No	Not described	No	Defined by a training cohort (unvalidated)
Hubers, 2014 A [36]	No	QMSP	No	No	No	No	Not described	No	Defined in a previous study
Hubers, 2014 B [29]	Yes	QMSP	Yes	Yes	No	Yes	Singleplex	Yes	Defined in a previous study
Hubers, 2015 [18]	Yes	QMSP	Yes	Yes	Yes	Yes	Singleplex Multiplex	No	Defined by a training cohort and validated in an independent cohort
Su, 2016 [20]	Yes	QMSP	Yes	Yes	No	No	Not described	No	Not described
Hubers, 2017 [37]	Yes	QMSP	Yes	Yes	Yes	Yes	Multiplex	Yes	Defined in a previous study
Hulbert, 2017 [28]	Yes	QMSP	Yes	Yes	Yes	Yes	Not described	No	Sensitivity and specificity values were obtained from the presence or absence of detectable methylation as a cutoff.
Su, 2018 [38]	Yes	Digital PCR	Yes	Yes	Yes	Yes	Not described	Yes	Defined by a training cohort and validated in an independent cohort
Li, 2021 [30]	Yes	Digital PCR	Yes	N/A	No	Yes	Not described	Yes	Defined in a previous study

## Data Availability

The data presented in this study are available in this article.

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
