# Peer review of "Methylated Cell-Free Tumor DNA in Sputum as a Tool for Diagnosing Lung Cancer—A Systematic Review and Meta-Analysis"

_cancers, 2024, doi:10.3390/cancers16030506_

Round 1

Reviewer 1 Report

Comments and Suggestions for Authors

A systematic review and meta-analysis was submitted for review using Embase, Medline, Web of Science and the Cochrane Library using the search strings: lung cancer, sputum and methylated tumor DNA. A total of 15 studies that met the selection criteria were included. The studies predominantly used a case-control design, with sensitivity ranging from 10 to 93% and specificity ranging from 8 to 100%. A meta-analysis of all genes across studies showed a pooled sensitivity of 54.3% (95% CI 49.4–59.2%) and specificity of 79.7% (95% CI 75.0–83.7%). Notably, two less studied genes (TAC1, SOX17) demonstrated sensitivity levels exceeding 85%.

1. I would like to see more details in Table 2: not just answers to yes/no questions, but also methodological details. What do the studies have in common: DNA extraction kits, the amount of DNA obtained, whether the quality of the obtained DNA was checked, etc. Can all this data be trusted, since the spread of values is significant?

2. Did these studies identify differences between squamous cell carcinoma of the lung and adenocarcinoma? stage of the disease, tumor location, etc.? Does the fact that the cancer is central or peripheral affect the amount of tumor DNA in sputum and the accuracy of diagnosis?

3. Figure 2 is of poor quality, needs to be redone, very pale and small inscriptions.

Author Response

Thank you very much for taking the time to thoroughly review and comment on our manuscript, we very much appreciate your time and effort. You have raised some important issues, which we have tried to address below: 

  1. Details in Table 2:

We have included the requested information on DNA extraction kits, amount of DNA used for the analysis, and the quality control of the DNA in Supplementary Table S1. However, most of the studies did not adequately report on the quality check of the DNA. As discussed, we believe that differences in the pre-analytical handling contribute to the variations observed among studies.

  1. Specificity according to stage, histology and location:

The studies did not report sensitivity/specificity in relation to stage, histology, or peripheral/central location. We have highlighted this in the fourth paragraph of the discussion.

  1. Quality of Figure 2:

As requested by the reviewer, we have enhanced Figure 2 by increasing the text size and changing the color from grey to black to improve readability. As described below (reviewer 3), the figure has been separated into three distinct figures.

Reviewer 2 Report

Comments and Suggestions for Authors

In this manuscript, the authors presented the current state knowledge on the potential of methylated tumor DNA in sputum as diagnostic maker for lung cancer, and conducted a meta-analysis to evaluate the sensitivity and specificity of all genes reported in the published studies as a biomarker for lung cancer detection. the authors extracted the quantitative methylated tumor DNA analysis data from 15 publish studies and performed the meta-analysis to evaluate the potential of methylated DNA analysis in sputum for the diagnosis of lung cancer. Identification of new biomarker for early diagnosis of lung cancer is of utmost importance. Through the meta-analysis of all genes investigated in early studies, the authors managed to identify that two less frequently investigated genes, SOX17 and TAC1 showed a high diagnostic sensitivity for lung cancer detection.

The topic and contents of this review paper are relevant in the field because identification of sensitive and specific biomarkers for early lung cancer detection is of utmost importance.

Although there were other published papers on methylated DNA as biomarkers for lung cancer detection, this review included updated information as it was based on the thorough literature search with references extracted on Oct. 3, 2023. It also included a meta-analysis, through which two less frequently investigated genes, SOX17 and TAC1, were found to have a high diagnostic sensitivity.

The conclusions consistent with the evidence and arguments presented and they address the main question posed. All of the references and Tables and figures are appropriate

Overall, the manuscript was well written. I don't have any concern about the data analysis and conclusions. 

Author Response

Thank you very much for taking the time to thoroughly review and comment on our manuscript, we very much appreciate your time and effort. We hope that this systematic review can help move the use of methylated tumor DNA towards clinical use. Thank you for helping us do that. 

Reviewer 3 Report

Comments and Suggestions for Authors

Lung cancer ranks second in the frequency of diagnosis and first in mortality among all types of cancer. A major problem is that lung cancer is usually diagnosed at advanced stages making treatment difficult. In this regard, the development of new minimally invasive and non-invasive methods for diagnosing lung cancer is of high value for modern medicine.

The use of sputum as a biospecimen for further analysis looks very promising. At the same time, quite a lot of attention has recently been paid to the diagnostic and prognostic potential of circulating tumor DNA, incl. level of ctDNA methylation. Thus, the systematic review and meta-analysis conducted by SWS Wen et al. is of great practical and scientific interest.

The review is well illustrated and contains tables that help to trace the history of the development of the problem and the progress of the analysis carried out by the authors. The supplementary materials presented strengthen the conclusions drawn by the authors. The authors took into account all possible meta-analysis biases and discussed their impact on the results in the relevant chapters.

The following are additional comments on the review by Wen et al:

The manuscript “Methylated cell free tumor DNA in sputum as a tool for diagnosing lung cancer – A systematic review and meta-analysis” is devoted to analyzing the possibility of using sputum as a biospecimen for diagnosis of lung cancer. To be specific, this review examines the possibility of using sputum extracellular tumor DNA methylation analysis in the diagnosis of lung cancer.

The topic of diagnosing various lung diseases by sputum often comes up in the scientific publications. The ability to diagnose infectious diseases is beyond doubt. However, the possibility of diagnosing lung cancer using sputum as a biospecimen is controversial. The presence of contradictions between the authors of original studies inhibits the continuation of scientific work on this topic. Unfortunately, funding in science is limited, so potential authors of new research tend to refuse risky research that does not guarantee a breakthrough result.

The systematic review with meta-analysis by Wen et al. can serve as a guide for other researchers involved in the development of diagnostic methods for lung cancer. Based on this review, it is possible to more accurately select the method of collecting sputum, paying attention, accordingly, to the preanalytical features of the preparation of biosamples, as well as shifting the focus of research from long-known genes that change the level of methylation in cancer (for example, RASSF1A) to less studied ones, for example, TAC1 and SOX17.

Over the recent 10 years, only 13 reviews devoted in more or less degree to the use of sputum as a bioassay and/or DNA methylation as a biomarker for lung cancer have been published. Undoubtedly, this is due to the small amount of original research in this area. However, every original study must be based on the analysis of already known data. The review by Wen et al. may provide such support.

At the last stage of selecting articles for review and meta-analysis, the authors excluded 52 of them. Reasons for exclusion were: a) wrong language - modern translation programs provide a sufficiently correct translation to include the original study, if not in the meta-analysis, then at least in the discussion of the results; b) wrong analysis method (for example, non-quantitative) - back in 2010-2012, many laboratories used both quantitative and non-quantitative methods to analyze DNA methylation levels, and the results obtained by qualitative (non-quantitative) methods were appropriate for good peer-reviewed journals. A brief review of non-quantitative studies of tumor DNA methylation in sputum could be included in a separate chapter of this systematic review. With other reasons for exclusion from the meta-analysis I would agree.

From my point of view, the conclusions are fully consistent with the arguments presented and meet the goals of the meta-analysis.

All references are appropriate.

However, if a decision is made to revise the manuscript and add a chapter on the analysis of tumor DNA methylation in sputum by non-quantitative methods, of course the list of references should be expanded. According to Figure 1, 26 publications were excluded based on wrong analysis method.

I have no comments on the tables. It seems to me that they contain all the necessary information; for details, it is better to refer to the original articles that the authors cite in the review.

Figure 2 is very cumbersome. In fact, this figure consists of several smaller figures and takes up two pages of the manuscript. In this form, it is poorly perceived by the reader. For me, it would be better to divide Figure 2 into three figures (for example, 2a and 2b; fig. 2c,d as 3a and 3b; fig. 2e,f as 4a and 4b) with a corresponding shift in the numbering of other figures. Then it will be easier for the reader to navigate the text and the figures illustrating the text. In addition, the labels for the axes in Figure 2 are grey, so they may be difficult to read when printing the article. It is better to make the labels for the axes in black.

The resolution of all figures is good, small details can be easily seen when enlarged.

After carefully review of the manuscript, I believe that it is worthy of publication, although some improvements could be made. Perhaps I would insist on dividing Figure 2 into several figures.  

Author Response

Thank you very much for taking the time to thoroughly review and comment on our manuscript, we very much appreciate your time and effort. Thank you for sharing your thought with us and for raising some very relevant points for us to consider. Please see below: 

  1. Inclusion of non-English studies:

Although translation programs have become available in recent years, these programs have not been validated for translating scientific papers. Hence, we believe that the inclusion of non-English studies would introduce substantial uncertainty about the results of the review.

  1. Inclusion of non-quantitative methodology:

The aim of this review is to gather the most recent and advanced evidence on methylated cell-free tumor DNA in sputum as a diagnostic tool for lung cancer. This involves employing the most sensitive methodology. Quantitative PCR is more sensitive than qualitative PCR, and including it, we believe, would compromise the results. The incorporation of qualitative PCR studies in a separate chapter is beyond the scope of the current review.

  1. Separation of Figure 2:

As requested by the reviewer we have separated Figure 2 into three distinct figures: Figure 2a and 2b; Figure 3a and 3b; Figure 4a and 4b. Furthermore, we have enhanced Figure 2 by increasing the text size and changing the color from grey to black to improve readability.

Round 2

Reviewer 1 Report

Comments and Suggestions for Authors

I have no further comments on the manuscript.